# A high-throughput multispectral imaging system for museum specimens

Wei-Ping Chan [1,2,10 ✉], Richard Rabideau Childers [1,2,10], Sorcha Ashe[1,10], Cheng-Chia Tsai [3], Caroline Elson[1], Kirsten J. Keleher [4,5], Rachel L. Hawkins Sipe[2], Crystal A. Maier [2], Andrei Sourakov[6], Lawrence F. Gall [7], Gary D. Bernard [8], Edward R. Soucy [9], Nanfang Yu[3] & Naomi E. Pierce [1,2 ✉]

We present an economical imaging system with integrated hardware and software to capture multispectral images of Lepidoptera with high efficiency. This method facilitates the comparison of colors and shapes among species at fine and broad taxonomic scales and may be adapted for other insect orders with greater three-dimensionality. Our system can image both the dorsal and ventral sides of pinned specimens. Together with our processing pipeline, the descriptive data can be used to systematically investigate multispectral colors and shapes based on full-wing reconstruction and a universally applicable ground plan that objectively quantifies wing patterns for species with different wing shapes (including tails) and venation systems. Basic morphological measurements, such as body length, thorax width, and antenna size are automatically generated. This system can increase exponentially the amount and quality of trait data extracted from museum specimens.

[1] Department of Organismic and Evolutionary Biology, Harvard University, Cambridge, MA, USA. [2] Museum of Comparative Zoology, Harvard University, Cambridge, MA, USA. [3] Department of Applied Physics and Applied Mathematics, Columbia University, New York, NY, USA. [4] Department of Forestry and Environmental Resources, North Carolina State University, Raleigh, NC, USA. [5] Department of Neurobiology and Behavior, Cornell University, Ithaca, NY, USA. [6] McGuire Center for Lepidoptera and Biodiversity, Florida Museum of Natural History, University of Florida, Gainesville, FL, USA. [7] Computer Systems Office & Division of Entomology, Peabody Museum of Natural History, Yale University, New Haven, CT, USA. [8] Department of Electrical and Computer Engineering, University of Washington, Seattle, WA, USA. [9] Center for Brain Science, Harvard University, Cambridge, MA, USA. [10]These authors contributed equally: Wei-Ping Chan, Richard Rabideau Childers, Sorcha Ashe. ✉email: chanw@g.harvard.edu; npierce@oeb.harvard.edu

Nanostructures in insect cuticles have inspired many novel engineering designs[1–5]. Since insects are known to be able to perceive wavelengths beyond the visible spectrum, important data can be missed unless the imaging systems used to survey insect cuticles are able to detect a full range of potentially relevant electromagnetic wavelengths. Current studies of lepidopteran (butterfly and moth) wing color (which in this paper we use as a shorthand for reflectance, agnostic of any visual system) and shape are often limited to less than 100 specimens[3,6] due to time-intensive single specimen-based procedures[7–9] such as the need to detach the wings from the specimens[2,4,10] or arrange and image individual specimens with their labels. Designing systems that accommodate wing shape diversity[9,11] has also presented a serious challenge.

The Lepidoptera provide an ideal imaging target since the two-dimensional nature of pinned specimens of most butterflies and many moths make them more tractable for analysis. Suitable methods are needed that can process multispectral images of Lepidoptera objectively, systematically, and efficiently. The main challenges are two-fold: (1) development of a high-throughput imaging system, and (2) identification of a universally applicable ground plan or archetype that can be generalized to capture wing characteristics across families.

Conventionally, the multispectral properties of the surface of an object can be measured in two ways[12–15]. A hyperspectral spectrophotometer provides high spectral resolution (~0.1 nm) for a single point, while multispectral imaging can quickly create two-dimensional images with high spatial resolution at some cost to spectral resolution by dividing the spectrum into multiple wavelength bands of ~100–200 nm each (hereafter referred to as "bands") and taking photo-like measurements over a large area using a camera. Some state-of-the-art imaging systems have 10-20 times finer spectral resolution (~5–10 nm), yet cost 70 times more than our apparatus (~$350,000). In remote sensing, satellites use multispectral imaging to collect data efficiently across large areas worldwide (e.g., Advanced very-high-resolution radiometer [AVHRR] and moderate resolution imaging spectroradiometer [MODIS]). Similarly, commercial multispectral cameras can provide objective multispectral measurements on two-dimensional surfaces, but those equipped with high spatial resolution are prohibitively expensive for most individual labs or museum collections and have relatively slow imaging efficiency, complicating their use in high-throughput specimen imaging. We therefore developed a scalable, high-throughput imaging system based on a modified consumer DSLR camera that can accommodate a Cornell style museum specimen drawer (450 × 390 × 67 mm) and is capable of collecting multispectral data from large numbers of biological specimens at once.

Another crucial component for comparing a taxonomically diverse set of insects is the use of an appropriate analytical framework that is robust to a range of morphologies. To date, the most common approach for comparing wing shapes and color patterns in Lepidoptera has been based largely on wing venation[9,16,17], which is highly correlated with both the evolutionary history and the physiological development of the species being studied. In his landmark study, Frederick Nijhout described a "general ground plan" to analyze pattern development in Nymphalidae and other butterflies based on variation in venation pattern and other morphological characteristics, guided by a relatively small set of developmental rules[18]. More recent evolutionary developmental approaches have turned away from analyzing shape and morphology per se and focused instead on identifying master regulatory genes (e.g., *optix*, *cortex Wnt A*) associated with wing color patterning, as well as the multiple cis-regulatory elements that enhance their effects[4,10,19–21].

While these advances have furthered our understanding of the developmental underpinnings of wing color patterns, they have not addressed the practical problem of how to make robust comparisons of widely variable wing morphologies. For example, the number of wing veins varies between different butterfly families[16,22], so it is not possible to employ a single venation system across butterflies. Wing shape also varies between species, with many Lycaenidae distinguished by hindwing tails that are not found in close relatives[23,24]. As a result, most studies of cross-family wing morphology have been carried out on forewings only, using wing metrics that affect flight, such as aspect ratio and moment of area[25,26]. For multispectral colors and patterns, many tools are available[27–29], but their application is largely limited to single clades sharing similar wing venation[2,9,28] or shape[7,28].

Biological museums preserve the wealth of the world's biological specimens, yet studies of wing color patterns to date have commonly focused on non-museum specimens because such research requires intact fore- and hindwings[2,4,10]. Additionally, imaging methods for these studies often require disarticulating specimens, limiting their utility for future research. Our imaging system overcomes some of these difficulties by non-destructively imaging whole pinned specimens across a customizable range of wavelength bands, and automatically processing them with an analytical framework that is robust to diverse morphologies across the Lepidoptera, including variable tail and wing shapes.

## Results

The two-dimensional nature of many pinned Lepidoptera specimens allows us to omit some technical considerations, such as the variable incident angles that need to be carefully considered when imaging 3D objects, and our multi-spectral imaging rig with its custom designed platform can image both the dorsal and ventral sides of specimens (Figs. 1a, 2 and 3). The initial descriptive data can be used for general multispectral property exploration and museum specimen digitization (Fig. 1b); the processed data, which are built on the initial descriptive data, can be used to investigate multispectral colors. After fore- and hindwing reconstruction (Fig. 1c), the design of the universally applicable analytical framework (Figs. 1d and 4a) can objectively quantify wing tails and accommodate different wing shapes and venation systems (Figs. 1e and 4b). The framework can also be applied to systematically survey multispectral color patterns (Figs. 1g, h and 5) while providing basic morphological measurements (Fig. 1f).

**The imaging system design.** Our high-throughput multispectral imaging system represents a compromise between the speed of traditional imaging and the need for objective spectral data. The system consists of a high-resolution SLR camera (Nikon D800) with its internal UV-IR filter removed to allow for UV–visible-IR imaging, fitted with a 28–80 mm f/3.3–5.6 G Autofocus Nikkor Zoom Lens (Methods). The customized imaging platform was designed to accommodate both ends of a pin, so mounted specimens can be positioned on the platform either dorsally or ventrally (Methods; Fig. 2). A reference bar containing black and white standard references and a scale bar is attached to the imaging platform in each round of imaging by a hook-and-loop fastener (Methods; Fig. 6a). The rough cost excluding the computational cost is ~$4500 (Supplementary Information).

For each set of specimens, a series of seven images (hereafter referred to as "drawer images"; Fig. 3a) are taken in raw format (*NEF) over the course of two minutes. These seven drawer images correspond to the following spectral imaging ranges (with details about light settings included in Methods): UV-only ($\lambda = 360$ nm; reflected light filtered through a Hoya U-340 UV-pass filter on the camera; combined UV reflectance and unfiltered

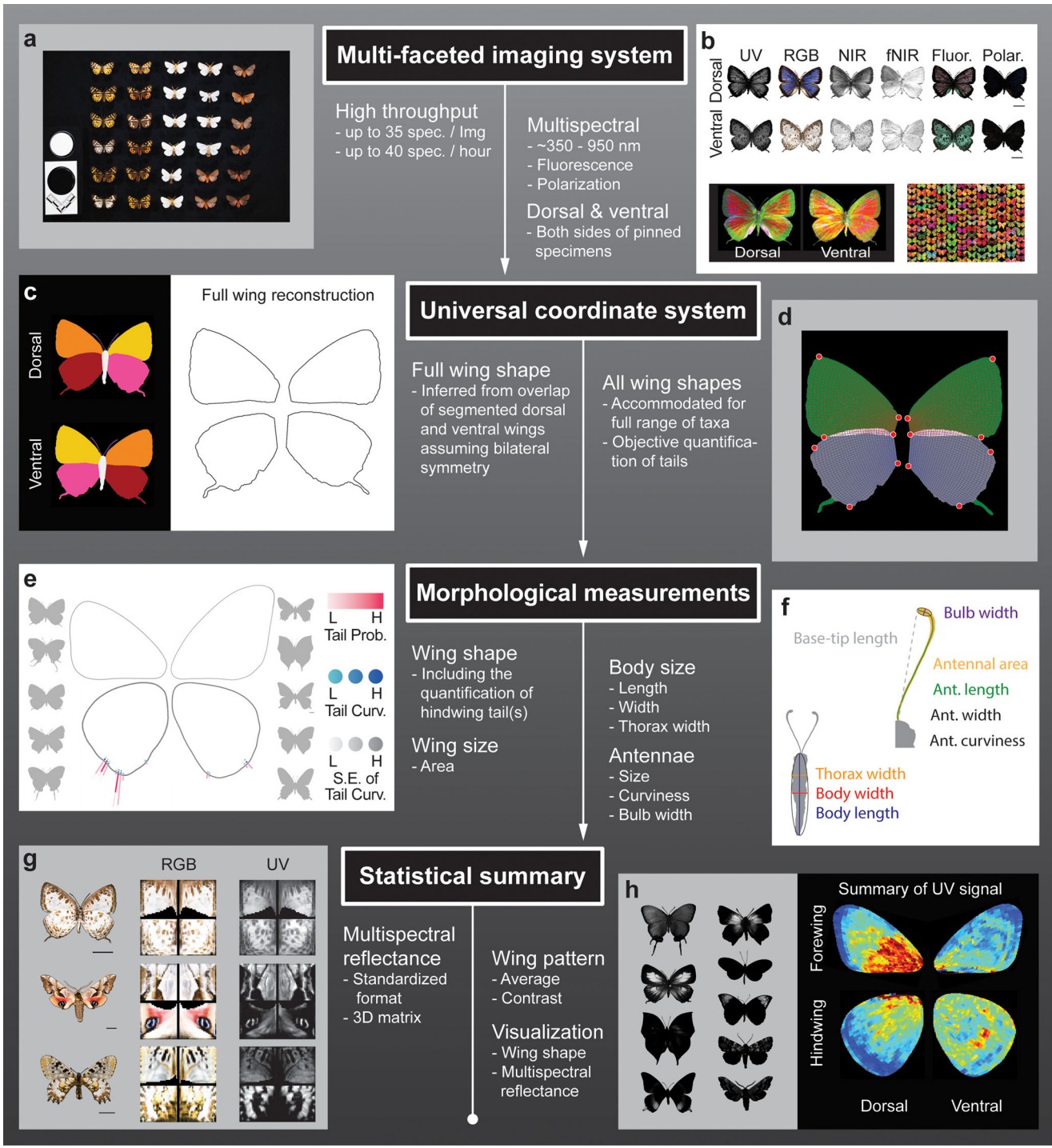

**Fig. 1 Summary of high-throughput, multi-faceted imaging methods.** The workflow runs from top to bottom, and the main features are highlighted. **a** An exemplar image showing the high throughput capacity of our imaging system. **b** Multispectral images (upper two rows) can be summarized as false color images (lower row) by principal component analysis, where red corresponds to PC1, green to PC2, and blue to PC3. **c** Full wing shape can be virtually reconstructed using information from dorsal and ventral segmentation. **d** A universal coordinate system for each wing can be generated automatically based on four landmarks (labeled as red dots). **e** Summarized wing shapes of two groups of butterflies: Lycaenidae on the left-hand side and Papilionidae and Nymphalidae on the right-hand side. Tail probabilities (Tail Prob.), curviness (Curv.), and the standard error of tail curvature (S.E. of Tail Curv.) are color coded accordingly. **f** Body and antennal morphologies can be measured automatically during image processing. **g** The summary of reflectance of four exemplar spectral bands (RGB and UV) from three specimens are shown. **h** Variation in UV reflectance of a group of butterflies is summarized as 'UV signal,' which represents the average contrasts of UV reflectance. Blue indicates low signal, and red indicates high.

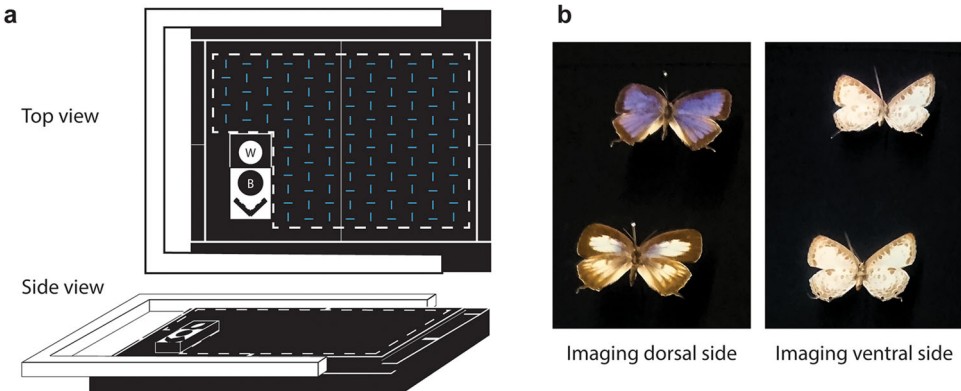

**Fig. 2 The design of the imaging platform accommodating both dorsal and ventral sides of specimens. a** The hidden pre-cut slots are illustrated in blue. White dashed lines, indicating the region that shows up in the image, were labeled by black sewn lines on the imaging platform for user guidance. The scale bar and black/white standard references are in the lower left corner. Details of the multi-layer foam backing can be found in Fig. 6. **b** The examples showing how dorsal and ventral sides of specimens being placed on the imaging platform.

visible fluorescence (called UVF hereafter) comprised of both reflected UV and all UV-induced visible fluorescence; two near-IR bands (unfiltered reflected light from λ = 740 nm [NIR] and 940 nm [fNIR] LEDs); and three in the visible (reflected broadband white LED light, λ = 400–700 nm, one unfiltered RGB and two RGB images filtered by linear polarizers at orthogonal angles to detect polarization along that axis), which are later decomposed into red, green and blue channels. Up to thirty-five pinned specimens can be imaged simultaneously, depending on their sizes, with wing sides facing either dorsally or ventrally.

**Drawer image processing**. All raw (multispectral) drawer images are uploaded to a high-performance computing environment, where we have developed a pipeline to process images automatically. However, a small number of images (<5) can be processed on a desktop with reasonable resources and longer runtime (Methods). To preserve the color gradient of the specimens, images are first converted into linearized 16-bit TIFF format[30] by dcraw[31] (an open-source program for handling raw image formats). These 16-bit TIFF images are then analyzed using MATLAB scripts. A set of seven drawer images is considered one computing unit, and the same group of specimens in the dorsal unit has a corresponding ventral unit (Methods).

Each computing unit (Fig. 3a) is read into memory, and the standard black and white references are recognized on the white (regular RGB) image by their circular shapes (Fig. 3b). Rather than calculating the exact number of absorbing photons at each sensor[13,27], we employ the remote sensing technique[32] of converting all pixel values into reflectance (albedo) units (between 0 and 1) according to the black and white reference standards (Methods; Fig. 3c). The scale on the drawer image is recognized automatically by local feature-matching to a reference image of the same scale, and the number of pixels per centimeter is derived (Methods; Fig. 3b).

Specimen pinning variability and optical aberration of the camera lens system would introduce measurement error during the imaging process. We estimate this error range in length measurement to be less than 0.4% (or 0.16 mm of a 4 cm butterfly (Methods; Fig. 7). Even though the error is minute, we leave a clear 5 cm margin around the edges of the stage when specimens are imaged in order to avoid relatively high aberration in the vicinity of the image boundaries (Fig. 7d).

Post-processing is applied to the UV, NIR (740 nm), fNIR (940 nm), and UVF bands to account for the differential sensor

sensitivity to these wavelengths in the red, green, and blue channels (Methods; Fig. 3c), except for the RGB-white band, which does not require post-processing. An index of polarization is calculated as the absolute difference between the two orthogonally polarized RGB white images. This single measure of polarization can also provide an indication of the occurrence of structure-induced colorations, suggesting whether additional studies should be carried out to investigate polarization at other viewing or incident light angles[33].

**Extracting specimen images from drawer images**. Our preliminary observations showed that Lepidoptera have the highest contrast with the background in the fNIR (940 nm) band, so we exploited this property to help recognize and extract individual specimen images from drawer images. (Methods; Fig. 3d). Each specimen's multi-band images were aligned into a layered image stack (Fig. 8a, b) based on affine geometric transformations, such as translation, rotation, scale, and shear. This step is relatively time-consuming, and processing time roughly scales with specimen size. At this stage, the registered multi-band specimen image stack, our initial descriptive data, can either be archived as part of a specimen's extended data or further transformed by our pipeline into higher-level processed data that produce shape, color and pattern trait data. For convenience, we included an additional binary mask layer with the information needed for background removal (Methods).

The completed initial descriptive data contain registered multi-band images (including UV, blue, green, red, NIR, fNIR, fluorescence [RGB], and polarization [RGB]), a background removal mask, and the scale bar) (Fig. 3a and 8a, b). Although further analysis is required to extract specific trait data from these datasets, they can be powerful visual aids in the discovery of novel wing scale types and structures. For example, the orange patches at the forewing tips of *Hebomoia glaucippe* (L.) show strong UV reflectance[14,15] (Figs. 3c and 8c), but the orange patches on *Chrysoritis pyramus* (Pennington) do not, suggesting a difference in the underlying physical mechanism producing these colors. Similarly, the white background on *Hebomoia glaucippe* shows little UV reflectance[14], but the white patches on *Arhopala wildei* Miskin (and many other species with white patches) show significant UV reflectance. (Fig. 8c). With a suitable converter, these initial descriptive data can be used in software packages[27,29] for analyses that take into account a range of animal visual systems. There is immense potential for discovery of multispectral

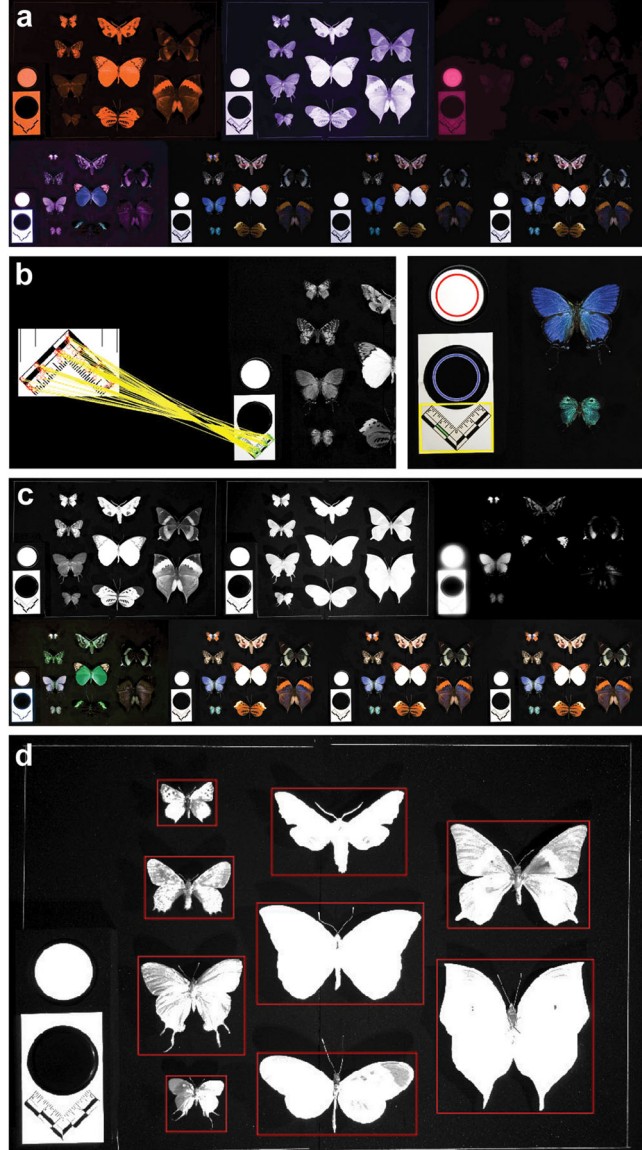

**Fig. 3 Imaging processing pipeline for each unit (a set of seven drawer images). a** A set of seven raw images in NEF format; **b** Automatic recognition of the scale bar and standard white and black references; **c** Reflectance calibration according to the standard white and black references; **d** Individual specimen images are extracted by red bounding boxes.

phenomena currently hidden within museum collections over the world for centuries by using these initial descriptive data alone.

A series of more complex analytical pipelines were designed to further quantify multispectral reflectance and shape traits. Following a detailed segmentation of different body parts, custom "tail quantification" and "wing-grid coordinate" pipelines are applied to record information about tails, wing shape, and multi-band reflectance traits.

**Body-part segmentation**. Our initial descriptive data include an overall specimen outline, but in order to segment this outline into different body parts, key landmarks are identified based on conventional geometry, including but not limited to mathematically searching the topology of the specimen outline (labeled as crosses and circles in Fig. 9b). We include two segmentation

methods. *Basic segmentation* (fully automated segmentation of specimen shapes according to landmarks with straight lines) can be used in the absence of data from the more time intensive *manual fore- and hindwing segmentation*. The manually defined fore- and hindwing segmentation pipeline is semi-automated, with human input through a stand-alone software package adapted from a GitHub repository named "moth-graphcut"[32], and the segmentations derived from it are more natural-looking (Fig. 9c). Basic segmentation is highly efficient, requiring no human input, but less accurate (inspection and correction are discussed later). In contrast, manual fore- and hindwing segmentation provides high accuracy of natural wing shape and full-wing reconstruction, with a throughput of approximately 100 specimen images processed per hour. In both methods, further morphological information, such as body size, body length, thorax width, antennal length, antennal width, and antennal curviness, are also automatically measured and collected along with the body-part segmentation (Methods; Fig. 1f).

Once specimens are segmented into body parts, the multi-spectral reflectance of each body part can be summarized (Fig. 9d). In addition to the analyses that can be done at the individual level with the initial descriptive data, more detailed comparisons can be made between the dorsal and ventral sides of different body parts. For example, by analyzing the reflectance of 17 specimens from 7 different families, we can observe that the dorsal hindwing shows significantly higher UV reflectance than its ventral side (Fig. 9e), possibly to assist in signaling, whereas the ventral side of the body and forewings shows higher fNIR reflectance than the dorsal side (Fig. 9e), possibly to assist in thermoregulation. However, additional processing is needed to produce coherent trait data within individual body parts that are comparable among more distantly related taxa.

**Universally applicable wing coordinates**. To compare multi-spectral wing traits across different wing shapes, we developed a generalizable pipeline consisting of four main components (Fig. 4): (1) complete wing shape reconstruction, (2) secondary landmark identification, (3) wing grid generation, and (4) hindwing tail summary. This system overcomes the particular difficulty of accounting for and quantifying diverse hindwing tails, and the processed data generated from this pipeline can also be directly applied in shape analyses.

In Lepidoptera and many other winged insects, a region of the hindwing often overlaps a portion of the forewing, complicating automated shape reconstruction. In our imaging paradigm, a specimen's hindwing is overlapped by the forewing in the dorsal-side image, and the forewing is overlapped by the hindwing in the ventral-side image (Fig. 4a). In our algorithm, the manually defined fore-hindwing boundaries are used to reconstruct the missing hindwing edge at the dorsal side and the incomplete forewing edge at the ventral side of a specimen. After reconstructing a complete wing, secondary landmarks are identified automatically (Fig. 1d and 4a). Tails on the hindwings are computationally separated from wing bodies before further processing (details about tail analyses can be found in Methods). A set of wing grids is then created according to the secondary landmarks of each wing (Fig. 1d and 4a). This grid system, which divides a specimen's silhouette according to the centroid of a set of four corners, is robust to the shape differences between different species, even for distantly related Lepidoptera (e.g., Sphingidae and Lycaenidae; Fig. 4b). Furthermore, the majority of these grids remain steady even in the presence of moderate wing damage (IV & VIII in Fig. 4b). The default resolution of these matrices is 32 × 32, but it can also be adjusted to accommodate specimens with larger wing areas.

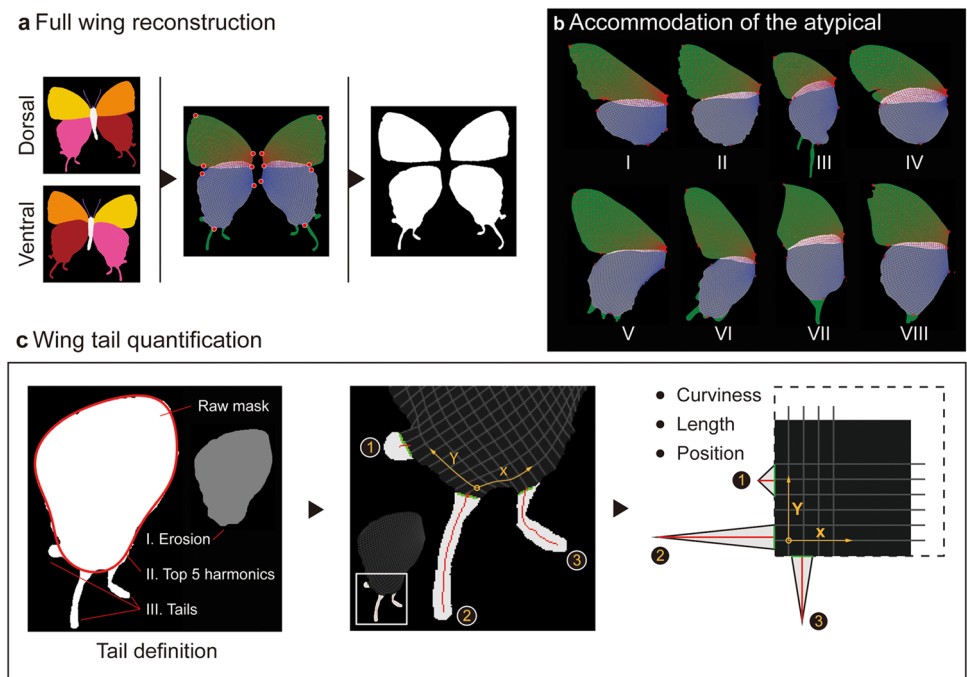

**Fig. 4 Diagram illustrating the full wing reconstruction, universal wing grids, and tail quantification. a** Full wing shape can be virtually reconstructed according to dorsal and ventral segmentation. **a, b** Wing grids can then be generated after **c** Defining the tail regions. In the left panel, the red boundary represents the reconstructed rough shape of a hindwing based on the top five harmonics after being projected into the frequency domain by elliptical Fourier analysis. **b** Universal wing grids can accommodate distorted or broken wings (e.g., IV & VIII). I. *Smerinthus cerisyi* (Sphingidae); II. *Catocala connubialis* (Erebidae); III. *Evenus coronata* (Lycaenidae); IV. *Heliconius melpomene* (Nymphalidae); V. *Allancastria cerisyi* (Papilionidae); VI. *Atrophaneura hector* (Papilionidae); VII. *Kallima inachus* (Nymphalidae); VIII. *Corades medeba* (Nymphalidae).

The quantification of hindwing tails and wing shapes also relies on this gridded system (Methods; Figs. 1e and 4c), and can be applied across the Lepidoptera (Fig.1e), without need for a priori identification of the presence or absence of tails. In contrast to other packages[28], our wing grid pipeline allows comparisons of diverse wing shapes, especially hindwings, with differing venation systems and tails (Fig. 4b). The even number of gridded anchors (e.g., 128 points in a $32 \times 32$ grid system) on the silhouette of a wing can be used as "landmarks" for shape comparison in other applications[9,11,34] (Fig. 4b). It can also be used to summarize multispectral wing patterns.

Based on this wing grid system, the average reflectance and variation of each grid can be calculated (Figs. 1g and 5a), and the results of a wing analysis can be stored in a $32 \times 32$ by N matrix (where N is the number of wavelength bands). The $32 \times 32$ resolution was determined by the size of the small specimens we handled; for example, it becomes meaningless to summarize data for a wing with $50 \times 50$ pixels using a finer resolution (e.g., $64 \times 64$). This standard format facilitates further statistical analyses among a wide variety of lepidopteran groups with different wing shapes.

The results of wing-patterning analyses can be further projected onto an average wing shape of a group for more intuitive interpretation (Figs. 1h and 5b). For example, the mean average reflectance identifies generally brighter wing regions (Fig. 5b) for RGB bands. High UV contrast regions appear to be important in UV intraspecific signaling[35], and we find that such regions are more likely to be seen on the dorsal side of Lycaenidae, but on the ventral side of Papilionidae (Fig. 5c, d). We can also compare the variability in the location of these high UV variable regions for a given group of taxa to show where they are highly conserved (low values) versus where they are more labile (high values; Fig. 5e, f). Such conserved regions indicate that UV variation (which could be involved in signaling) in that wing region (whether present or not) is highly constrained and therefore stable across different species. Although these are examples chosen to illustrate a wide variety of wing shapes rather than targeting a specific scientific question, they already begin to provide biological insights for further study, demonstrating the utility of carrying out systematic studies of lepidopteran traits using this approach.

**Inspection, manual correction, and visualization**. Given the relatively large file sizes (~240 Mb per image) and time intensive post-processing pipelines, most of our protocols are designed to be run in high-performance computing environments (i.e., clusters); however, inspecting and manually correcting the images are inconvenient in such environments. We therefore designed the pipeline to enable a small proportion of the dataset to be downloaded to a local machine for inspection and manual correction. In total, our pipeline has five potential points where inspection and manual correction are possible (Methods). At each inspection point, we also developed corresponding scripts and user interfaces to manually correct the dataset on local machines with minimal resource requirements (low storage, memory, and CPU requirement). Scripts for customized visualization settings were also developed for wing shape (including tails) and wing patterns (Methods, Supplementary Information, and Data availability).

## Discussion

This system allows researchers to efficiently produce and archive high-quality and informative multispectral images of museum specimens. Having applied it to more than 10,000 specimens to date, we have found that a Cornell drawer of specimens (~60–80 individuals) can be imaged in 2 hours given our current workflow. This estimate includes specimen handling, retrieval, and re-

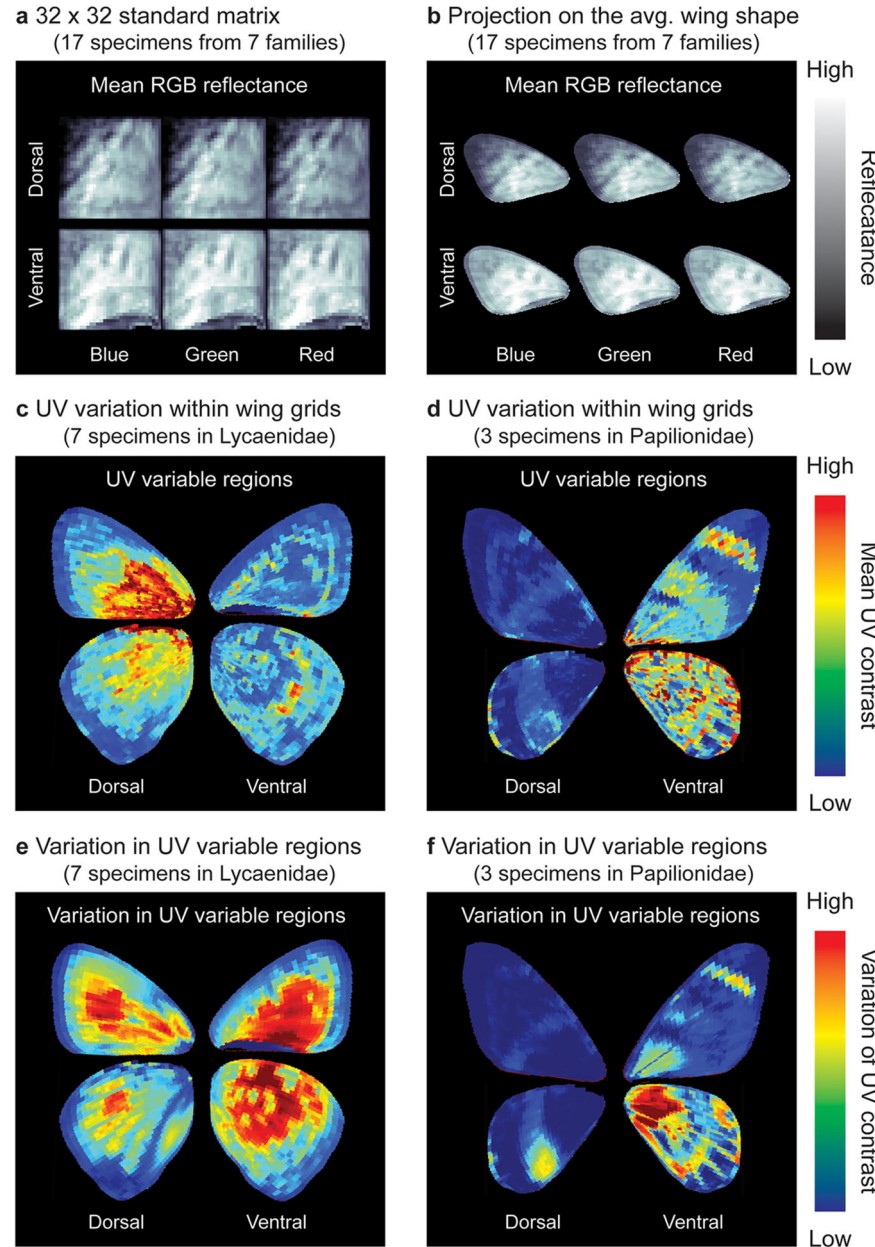

**Fig. 5 Reflectance summary from exemplar bands according to wing grids. a** Summarized gridded reflectances in blue, green, and red bands can be **b** Projected onto the average wing shapes of a selected group of specimens. **c**, **d** UV signal (the average UV contrast among species) showing the most common likely highly UV variable regions on the wings. **e**, **f** Variation of UV variable regions between species, indicating regions of UV patterning that are highly conserved (blue) versus those that are more actively changing (red).

integration into the collections, although the imaging time typically scales inversely with the size of the specimens. Digitizing museum collections has become a mission of institutions worldwide, with vast numbers of specimens processed in the past decades[36,37]. These digitalized records were applied for scientific and social purposes with support from governments and citizens through different curating systems and platforms, such as GBIF (http://www.gbif.org), iDigBio (http://www.idigbio.org), MCZBase (https://mcz.harvard.edu/database), Atlas of Living Australia (http://www.ala.org.au/), Map of life (https://mol.org/), and ButterflyNet (http://www.butterflynet.org/). Imaging systems and pipelines, ranging from 2D traditional photography[37] to 3D CT-scan[38,39] and 3D photogrammetry[39,40] with or without convenient user interfaces, have also seen vast improvements, often with correspondingly vast prices. Our multispectral imaging system provides an important step forward for those interested in

high-throughput spectral phenotyping or digitization of specimens in a cost-effective manner, which we hope to continue to adapt as the field of archive digitization matures.

We anticipate a number of potential improvements to the current system. On the hardware end, incorporating a rotational plane into our current imaging platform will allow researchers to study reflectance at different incident angles[33], which we initially did not include due to the two-dimensional nature of most lepidopteran specimens. Wings that have been detached from the body of an insect cannot be accommodated by our current pipeline, so the development of an imaging platform to mount individual wings will also enlarge the potential utility of this system.

On the software end, efficiency would be greatly improved by reducing the need for manual input[41]. For example, with sufficiently large numbers of previously processed images as a training set, an automatically inspecting and self-correcting system can be

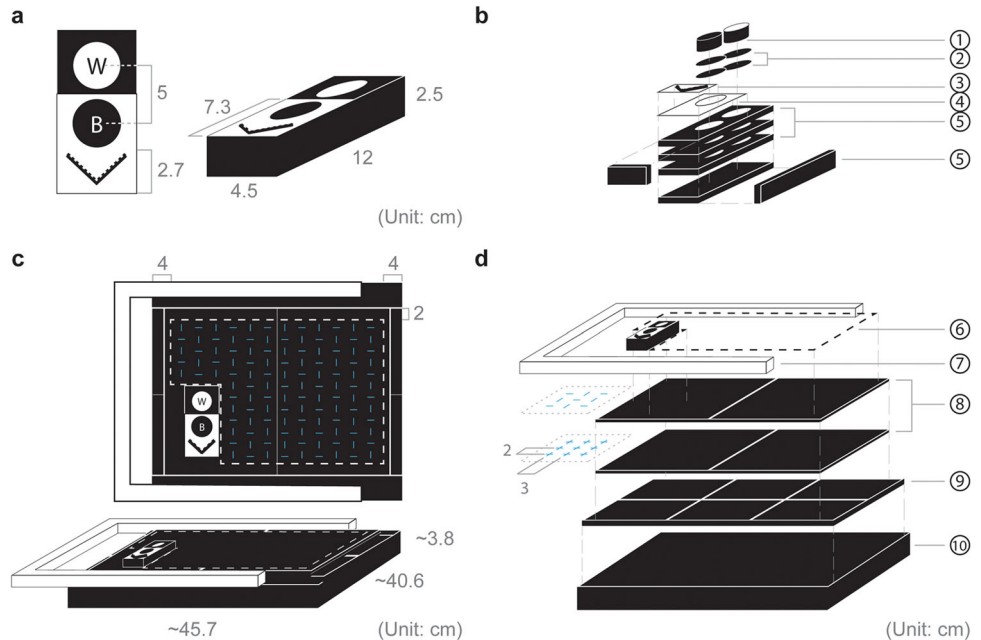

**Fig. 6 The design of the imaging platform. a** Reference bar and **b** Corresponding materials. **c** Imaging platform with the hidden pre-cut slots shown in blue and **d** Corresponding materials. (1) Black (spectralon black standard AS-001160-960, Labsphere) and white (spectralon white standard AS-001160-060, Labsphere) standard references; (2) Hook loop strips with adhesive (2.4″ × 2.4″); (3) Adhesive forensic evidence labels (2 cm measure; A-6243); (4) Adhesive tear resistant waterproof photo craft paper; (5) Neoprene sponge black foam pads with adhesive (12″ × 8″ × 1/4″); (6) Black cotton thread; (7) Canvas stretcher bar kit (16″ × 20″); (8) Neoprene sponge black rubber foam with adhesive (12″ × 8″ × 1/8″); (9) Neoprene foam anti vibration pads with adhesive (6″ × 6″ × 1/4″); (10) Polyethylene foam (18″ × 16″ × 1.5″). Detailed information can be found in Supplementary Information.

developed based on machine learning. A similar approach could be applied in the case of fore-hindwing segmentation. The utility of this imaging protocol in large-scale morphological studies of Lepidoptera will be further advanced if more detailed body part segmentation (e.g., eyes, legs, and proboscis) could also be developed.

The current version supports only those specimens with heads positioned toward the top of an image. An automatic rotational correction could help orient specimens for optimal downstream processing. The user interface could also be improved, as the current design requires back-and-forth communication between a computing cluster and the user's local machine. A unified user interface could help to simplify operation of the complex protocols.

To connect the results of the image analyses described here with existing knowledge in the fields of evolution and developmental biology, integration of the data with evolutionary insights from wing venation systems seems essential. The wing grid system provides universally applicable coordinates for shape and reflectance comparison across diverse lepidopteran taxa. By registering wing venation systems on these wing grids, the relationships between venation, multispectral reflectance, and shapes can be further explored.

Museum collections are vast repositories of biological information of both basic and applied value, simply waiting to be mined. The relatively inexpensive and user-friendly imaging hardware and wing grid processing software presented here will enable museum researchers to explore with high efficiency the multispectral properties of not only Lepidoptera but also many other insect groups. It will also facilitate the comparison of colors and shapes among species with highly diverse wing shapes, in contrast to other available packages for the study of colors and patterns. These methods can be easily adapted to study other similarly two-dimensional subjects, such as the leaves of plants or cultured microorganisms. Our methods have the potential to revolutionize the efficiency and accessibility of collecting reflectance and shape data for biological specimens, providing a rich source of information for bio-innovation from collections worldwide.

## Methods

**The imaging system design**. The system consists of a high-resolution SLR camera (Nikon D800), fitted with a 28–80 mm f/3.3–5.6 G Autofocus Nikkor Zoom Lens. The camera is mounted inside a rectangular light box constructed of 0.125″ thick 6061 Aluminum sheeting (McMaster-Carr: 89015K18), mounted on T-slotted aluminum framing extrusion (McMaster-Carr: 47065T101), which is 36 inches tall, and 24 inches wide and deep, and open at the bottom. Inside the light box, 4 banks of LED emitters are mounted 18 inches high on the sides, on thick aluminum heat sinks which can be rotated up and down to provide direct or indirect illumination. Each bank of LEDs is composed of 4 star metal-core-printed-circuit-boards (MCPCBs, OSRAM Opto Semiconductors Inc.), one for each wavelength band, and each with 6 individual LEDs. The four wavelength bands are ultraviolet (UV 365 nm: LZ1-30UV00-0000), white (Cool white: LZ1-10CW02-0065), 740 nm IR (740 nm Red: LZ4-40R308-0000), and 940 nm IR (940 nm Red: LZ1-10R702-0000). The camera is screw-mounted to a monopod (Sinvitron Q-555) attached to a piece of framing extrusion extending through the middle of the light box, with the lens held 28.25 inches from the bottom. A motorized filter wheel with four slots is mounted directly underneath the lens, with an empty slot for unfiltered RGB white, UVF, NIR, and fNIR imaging, a Hoya U-340 UV pass filter for UV-only images, and two B + W KSM circular polarizers mounted at orthogonal angles, for differential white polarized imaging.

A microcontroller (PJRC, Teensy ++ 2.0) and stepper driver (Sparkfun, ROB-12779) control a motor (Mercury Motor, SM- 42BYG011-25) that rotates a bespoke filter wheel referenced to an origin home switch. LEDs are driven by banks of current controllers (LEDdynamics, 3021-D-I-100) with a microcontroller coordinating illumination timing (PJRC, Teensy 3.2). The camera is controlled by the open-source DigiCamControl software (DigiCamControl V2.0.0). Coordination of camera operation, LED illumination, filter wheel positioning and image transfer is done by a desktop computer running a custom LabView program[42]. All software and hardware designs are available upon request.

To minimize background reflectance, the material used in building the platform was carefully chosen and tested for spectral neutrality from the ultraviolet through the near-infrared bands (Fig. 1a and 6). We included a series of pre-cut slots in the underlying multi-layer foam backing so that pinned specimens could be easily pushed in and held by either the top or the tip of the pin, to enable efficient dorsal and ventral imaging (Fig. 2b). A reference bar containing black (spectralon 2% reflectance AS-001160-960, Labsphere) and white (spectralon 99% reflectance AS-

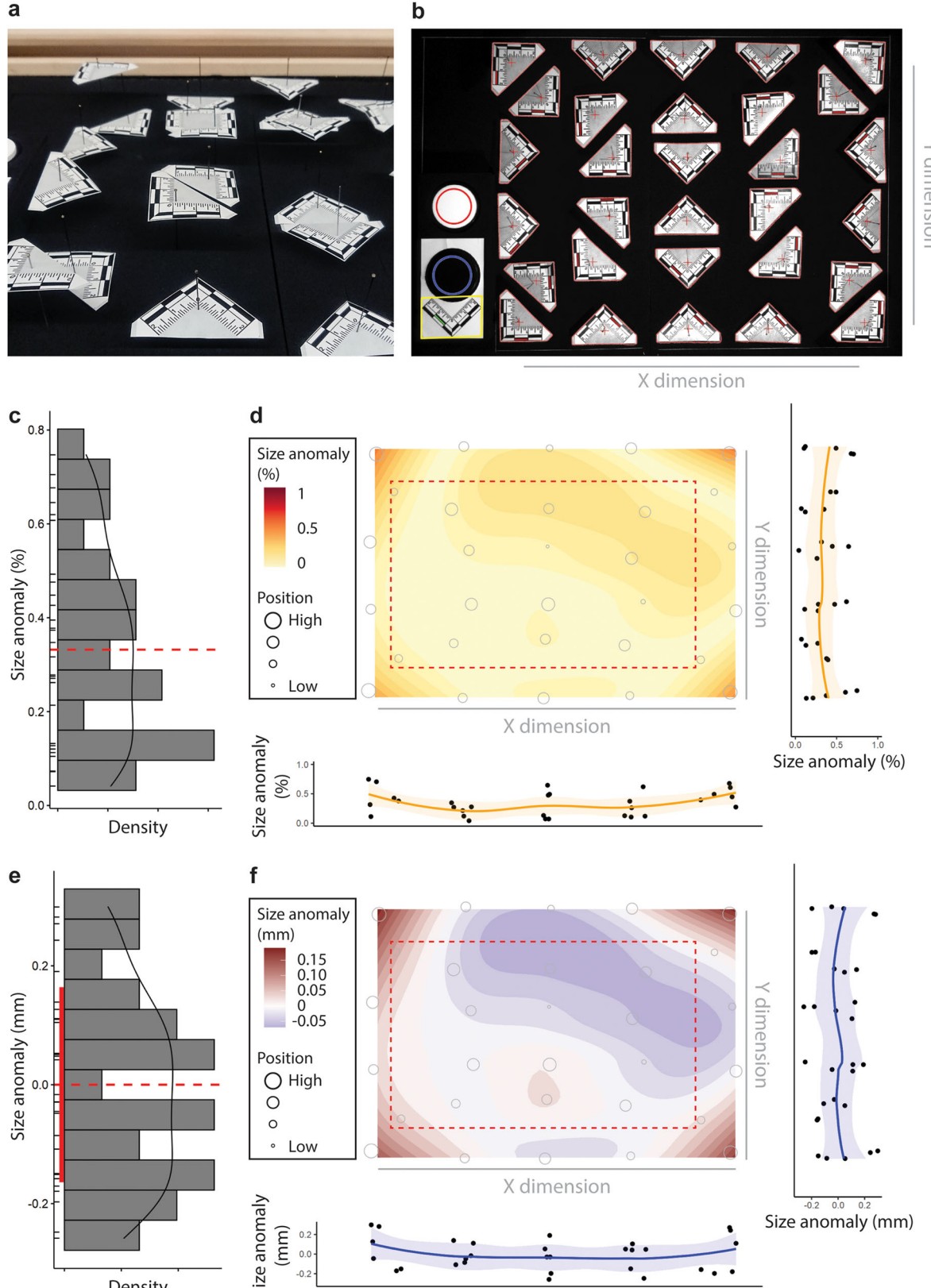

**Fig. 7 Empirically measured lens-induced measurement distortion. a** Illustration of scale standards placed at different heights on pins. **b** View of the image taken by the imaging system. **c, d** Frequency distribution **c** and spatial distribution **d** of size anomalies in absolute percentage compared to a 4 cm butterfly. **e, f** The raw value for **d** & **e** is provided. The dashed line in **c** & **e** indicates median and zero, respectively. The region bounded by the red dashed line in **d**, **f** are where we place our specimens for imaging.

**a** Image layers of one side of a specimen

**b** The view of each layer

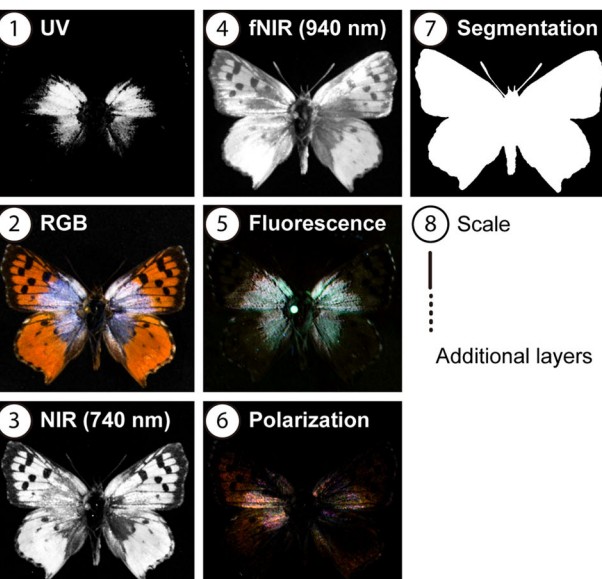

**c** Special multispectral properties

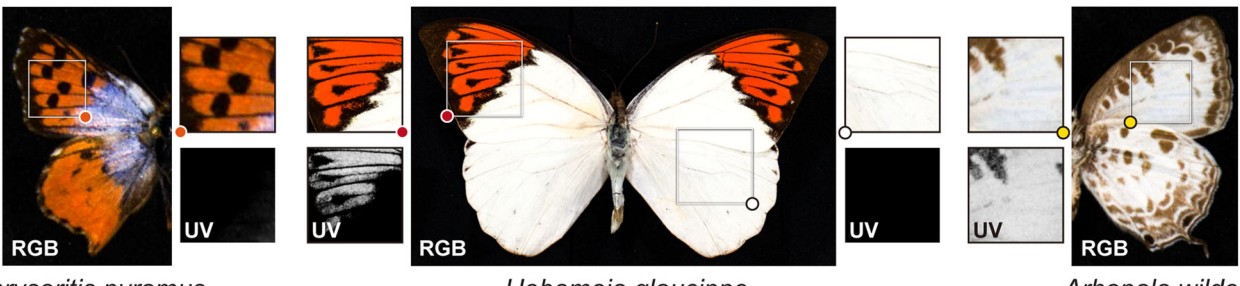

*Chrysoritis pyramus*                     *Hebomoia glaucippe*                     *Arhopala wildei*

**Fig. 8 The data structure of the initial descriptive data, with preliminary multispectral insights. a** The multi-layer format ("cell format", the technical term used in MATLAB) is applied to contain the descriptive data for a single side of a specimen. **b** The appearance of some exemplar layers for the South African lycaenid, *Chrysoritis pyramus*. The actual output layer order can be found in the Supplementary Information. **c** Variation in scale properties affecting UV reflectance can be found within a single specimen of *Hebomoia glaucippe*, compared here with *C. pyramus* on the left and *A. wildei* on the right.

001160-060, Labsphere) standard references in contrasting background colors and a scale is attached (Fig. 6). We tested the system using Lepidoptera specimens from the Museum of Comparative Zoology Entomology Collection. The imaging surface of each specimen was adjusted to approximately the same height as the standard reference bar.

**Drawer image processing**. A set of seven drawer images is considered one computing unit, and the same group of specimens in the dorsal unit has a corresponding ventral unit. Each unit is independently processed such that all units can be processed in parallel on the cluster. The resources allotted for each job/unit are set to two cores with twenty-four-gigabyte memory for twelve hours. Most images can be processed fully within six hours, but larger specimens (e.g., Papilionidae, Nymphalidae, and Saturniidae) take longer. Here, multiple approaches were chosen to maximize the degree of automation for highly varied specimen conditions, increasing the image processing time from minutes to hours. While simple algorithms may be efficient, they are not generalizable. Under faster but more simplistic conditions, the number of outliers that would need to be manually handled would increase, ultimately consuming more time and human labor than that spent on computation in a high-throughput situation.

For each unit, a set of seven drawer images (Fig. 3a) is processed after the standard black and white references are recognized (Fig. 3b). The reflectance of a circular patch at the center of each standard can be extracted for all bands (avoiding the margins, which can become more easily distorted or contaminated accidentally as a byproduct of frequent imaging) (Fig. 3b). By comparing the imaged pixel intensity with the known reflectance values of the standards for calibration, we can rescale and calibrate all pixels in the image[13,30] (Fig. 3c) with the reflectances of the standard references provided by the vendor (Supplementary Information). Values can differ slightly from one standard to another, so they each

need to be measured independently to provide an initial baseline. The scale on the drawer image can be recognized automatically by local feature-matching with a given reference image of the same scale. Feature points from the two images (the reference and the drawer image) can be extracted, and the matching points identified by the speeded up robust features (SURF) descriptor[43], which is an advanced version of the scale-invariant feature transform (SIFT)[44]. Further conventional image processing procedures (e.g., erosion, dilation, and object filtering) are then applied to the detected scale in order to derive the number of pixels represented in one centimeter (Fig. 3b).

The post-processing of each of the remaining bands is as follows: due to the non-overlapping mosaic design of RGB Bayer filters, there are more green than red and blue light receptors in consumer SLR cameras[13,30,45]. Under low RGB light environments, the signals received by green sensors are thus more likely to be used to estimate the missing color values, which compensates for insufficient signals detected in red and blue sensors. This phenomenon was manifested in our UV images, so the green channel was excluded whenever a UV image was calibrated. For NIR (740 nm), which is not far from the detected spectral range of a camera's blue sensors, the blue channel was not included in deriving the NIR (740 nm) product, because the camera's blue sensors may still be able to detect minute NIR signals and thus introduce noise. In contrast, fNIR (940 nm) is more distant from the detection of blue sensors, so the normal RGB calibration was applied. For the fluorescence in all RGB channels, we calculated the reflectance difference between UVF and UV images (UVF deducts UV). We did not avoid the green channel of the UV images in this case, or we would not have obtained reasonable intensity values for green fluorescence. The fluorescence quantified by our approach should only be compared with objects measured using a similar approach. Since fluorescence and some wavelength bands (e.g. UV and polarization) are typically dim compared with other wavelengths, the images shown in this paper have been adjusted for better human visibility.

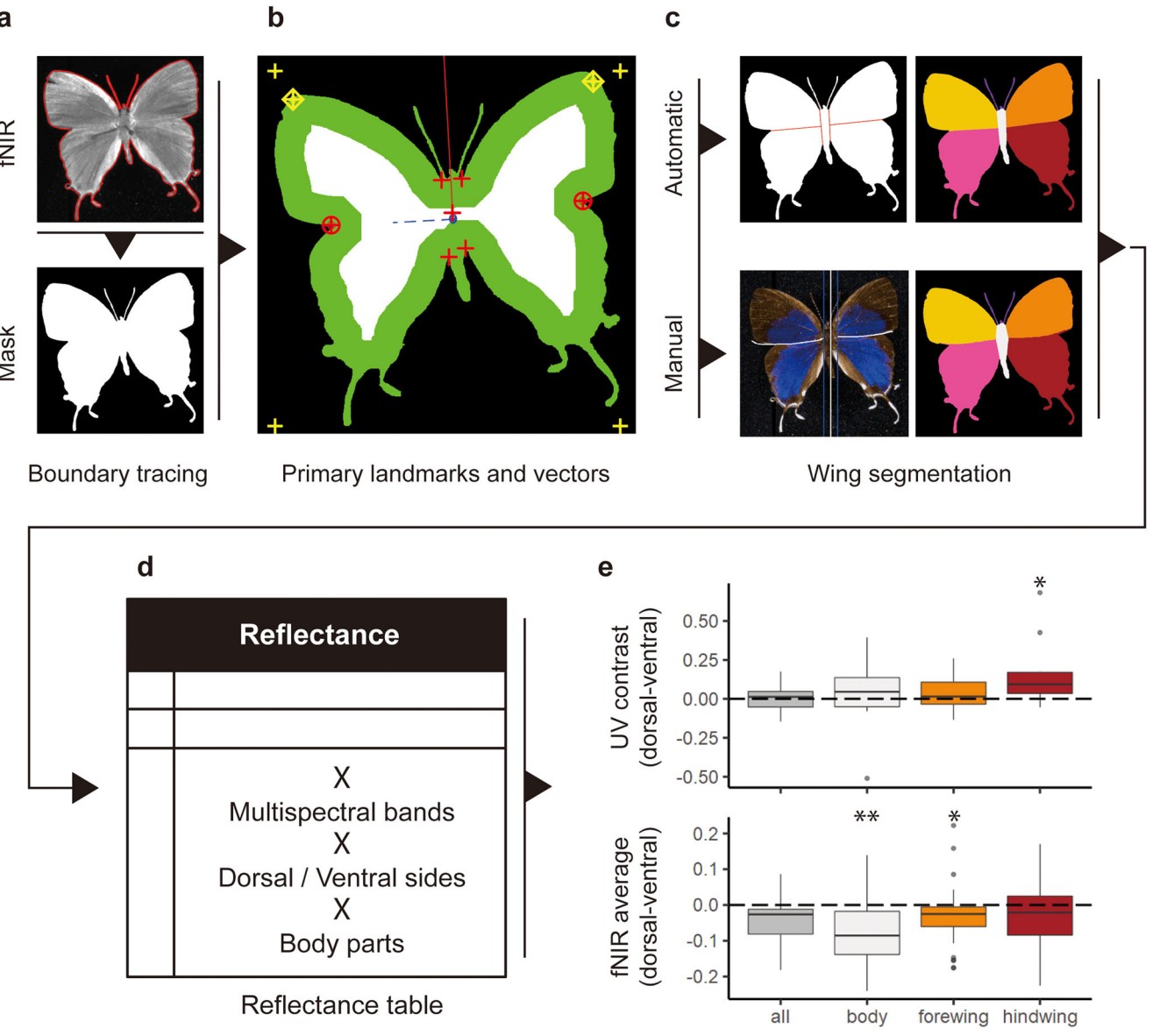

**Fig. 9 Details of body part segmentation with preliminary observations. a** Background removal based on fNIR image. **b** With primary landmarks (represented by crosses and circles) and vectors identifying the symmetrical axes (represented by segments in the solid red line and the blue dashed line), the specimen mask can be automatically segmented **c** With or without manually defined fore-hindwing division of the overlap region. The species shown in (**a–c**) is *Oxylides faunus*. **d** Statistical summary (mean, variation, and patch size) of all bands for both sides (dorsal and ventral) of all body parts (four wings and the body) can be calculated. **e** Statistical results of exemplar bands based on 17 specimens from 7 different families. Center line represents median; box limits, upper and lower quartiles; whiskers, 1.5x interquartile range; points, outliers. Statistically significant levels of difference between dorsal and ventral sides (under two-tailed t-test) are labeled by asterisks: *, 0.05; **. 0.01.

**Validation of size measured in the system**. We use conventional lenses in our imaging system for cost-efficiency and user friendliness, but such lenses are prone to spatial and optical distortions, usually towards the boundaries of the image. In order to quantify the magnitude of these distortions on the measurements taken by our imaging system, thirty scale bars were pinned at different heights (to simulate extreme variability in pinning height) and spread on the imaging platform (Fig. 7a, b). Different pinned heights were included for each column and row, although this resulted in the scale-bars at the four corners all being near the top of the pin. The RGB image (Fig. 7b) was then analyzed in the image processing pipeline. For each scale bar, the location on the imaging platform and the average number of pixels indicating one centimeter were recorded.

**Excising specimen images from drawer images**. Given that our approach was designed to accommodate the high diversity in Lepidoptera, specimens with diverse wing shapes can be imaged together (Fig. 3). The rough position of each specimen is determined according to the fNIR image, and a rectangular bounding box is generated with the inclusion of buffer zones on all four edges (1/5 specimen height

to the top; 1/15 height to the bottom; 1/20 width to the left and right boundaries). Specimens are cropped according to the corresponding bounding boxes (Fig. 3d), and difficult targets (e.g., legs and antennae, and stains formed by fallen scales) on the imaging platform are automatically filtered out by the cropping algorithm. This function may accidentally remove tiny specimens, so the threshold value is designed to be specified manually according to the minimal specimen size to be imaged on the imaging platform. Efforts to automate this step were not feasible. If the filtering procedure is automated, small specimens will be automatically filtered out as a non-specimen object when large and small specimens are imaged together. For example, the size of a fallen papilionid body part can be as large as the size of a small lycaenid with one missing wing, and we would want to filter out the former but retain the latter.

Background removal, where a specimen is selectively cropped from the image background, is part of the segmentation process and involves many steps, so only the broad outline of the procedure is provided here. Since the intensity of reflectance differs from one species to another, a consensus approach based on three segmentation techniques was used to handle the diverse range of Lepidoptera: k-means clustering[28,46], Gaussian filtering[29,47], and active contouring[48,49]. The

K-means clustering technique divides a false-color RGB image composed of one NIR and two fNIR bands, into five clusters (k = 5). The background color and location are then identified, and the remaining regions are labeled as the specimen. The Gaussian filter technique uses a Gaussian filter on the entire image to smooth out relatively minor variations within a specimen while maintaining high contrast between the boundaries of a specimen and the background, facilitating conventional segmentation techniques. The active contouring technique (a.k.a. Snakes) was also applied to find an objective outline of a specimen by growing iteratively from the initial specified region towards object boundaries (Fig. 9a).

Antennae and abdomens sometimes require additional attention during the segmentation stage, particularly when they overlap or touch other body parts being segmented. Legs extending out from under the abdomen can interfere with accurate hindwing cropping. Using simple image erosion (deduction with some basic logic), antennae touching the leading edge of the forewing are preserved as much as possible without damaging the forewing shape. However, legs that intersect other body parts are automatically removed to preserve hindwing shapes. In rare cases, rectangle bounding masks were generated as placeholder masks when the specimen could not be excised from the background in one piece or when serious errors occurred in the process of forming specimen masks. Detailed examples and descriptions can be found in the Supplementary Information.

Specimen barcodes were used as the file name for a set of specimen images (dorsal and ventral sides). A dataset (in csv format) containing the information of all image names and imaged specimens, which can be generated manually, was required at the image processing stage (find Protocol in Supplementary Information); otherwise, temporary barcodes (e.g., "Tmp-1" and "Tmp-2") were automatically assigned to name a set of specimen images.

**Further morphological information**. Information regarding body size, body length, and thorax width are measured after the virtual removal of the four wings (Fig. 1f). For an antenna, a series of measurements were developed (Fig. 1f): the length of a path tracked along a curved antennal mask corresponds to antennal length; the average width of an antenna can be derived from the mask area of an antenna divided by its length; and antennal curviness is calculated as antennal length divided by the direct linear distance between its tip and base. The size of an antennal bulb can also be obtained from the width of the tip of an antenna. These morphologies were also quantified for a broken antenna, so in the application of antennal traits, one should carefully filter out the data of broken antenna manually or systematically. To reasonably compare this trait among different individuals, we suggest using the ratio between the size of antennal bulb and the overall antennal width as the meaningful comparable quantification.

**Tail quantification**. A general erosion of N pixels (which is scaled by the size of the specimen algorithmically; a high-frequency value is 5 in our imaged specimens) was first applied to remove tiny silhouette features created by hairs and attachments (e.g., crystalized chemicals and large dust grains). The outline of the specimen mask is then projected into the frequency domain by elliptical Fourier analysis[50], and the top five harmonics are used to reconstruct the rough shape of a hindwing. The areas extended from these reconstructed wing regions are defined as tails (Fig. 4c). The morphology (length and curvature) of those independent areas can be further quantified and recorded according to the wing grid system (Fig. 4c).

**Inspection, manual correction, and visualization**. In total, our pipeline has five potential points when inspection and manual correction are possible: (1) the bounding box; (2) the specimen mask; (3) the segmentation of fore- and hindwings; (4) the identification of primary landmarks; and (5) the application of wing grids. The module that generates images for inspection is embedded in the image-processing pipeline, so these images can be easily found in the specified result directories. Most manual correction tools for the local computer have been written in MATLAB, except for the specimen mask correction which requires commercial painting software (such as Adobe Photoshop) and the fore- and hindwing segmentation task (written in Python). Corresponding scripts have also been prepared to update the dataset on the cluster with the manually corrected information.

**The advanced visualization system**. Many visualizations are automatically generated within the image processing pipeline. However, some species have special wing size and shapes, so more customized settings may be required for better visualization. A script developed for customized wing shape and tail visualization is also provided (Supplementary Information).

**Data structure**. The data structures of the initial descriptive data as well as the processed data and the group summary matrix are provided in Supplementary Information.

**Statistics and reproducibility**. By following the pipeline with the raw data provided in the Supplementary Information, all intermediate and final products are readily reproducible. The figures and figure legends give the sample sizes, number of specimens and species, as well as the applied statistical approach.

**Reporting summary**. Further information on research design is available in the Nature Portfolio Reporting Summary linked to this article.

## Data availability
The datasets generated and/or analyzed during the current study are available at DryAd (https://doi.org/10.5061/dryad.37pvmcvp5)[51].

## Code availability
Detailed step-by-step instructions are documented on Protocols.io with tutorial videos for some crucial steps. All source codes are provided at GitHub. Please find Supplementary Information for details.

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

## Acknowledgements

We thank Chen-Ming Yang for sharing the source code and advising us when we developed the tool for fore- and hindwing manual segmentation; Zoe Flores for testing all pipelines and scripts; Josh Sanes for his advice and encouragement through the Center for Brain Science at Harvard; and David Lohman for advice and assistance in identifying suitable personnel for data entry as well as for determining appropriate nomenclature. We thank Joel Greenwood for his help in the early development of the imaging system. We thank Sarah Maunsell, Jalen Winstanley, Amy Wu, Even Dankowicz, Clayton Ziemke, Christian Alessandro Perez, Ling Fang, Avalon Owens, Zhengyang Wang, Cong Liu, Katherine Angier, Evan Hoki, Francisco Matos, Beaziel Ombajen, Zoe Flores, Jocelyn Wang, Han-Ting Yang, Jingqian Wang, Jiale Chen, Annina Kennedy-Yoon, Atreyi Mukherji for testing and helping to optimize different protocols and tools for inspection and manual correction. W-PC was supported by a graduate fellowship from the Department of Organismic and Evolutionary Biology at Harvard University; SA was supported by a Herchel-Smith grant from Harvard University; RARC was supported by a Graduate Research Fellowship (GRFP) from the National Science Foundation (NSF), C-CT was supported by a graduate fellowship from the Department of Applied Physics and Mathematics at Columbia University. This research was supported by the Air Force Office of Scientific Research FA9550-14-1-0389 (Multidisciplinary University Research Initiative) and FA9550-16-1-0322 (Defense University Research Instrumentation Program) to NY, and by NSF PHY-1411445 to NY and NSF PHY-1411123 to NEP and NSF DEB-0447242 to NEP. Published with a grant for Open Access from the Wetmore Colles Fund of the Museum of Comparative Zoology.

## Author contributions

Conceptualization: R.R.C., C.-C.T., G.D.B., E.R.S., N.Y., N.E.P., Data collection: W.-P.C., S.A., C.E., K.J.K., R.R.C., E.R.S., Data curation: R.L.H.S., C.A.M., A.S., L.F.G., Hardware development: R.R.C., E.R.S., Software development: W.-P.C., Formal analysis: W.-P.C., Visualization: W.-P.C., Validation: W.-P.C., S.A., Writing—original draft: W.-P.C., R.R.C., S.A., N.E.P., Writing—review & editing: All authors

## Competing interests

The authors declare no competing interests.
