## [Peer Review File · Communications Biology]

Reviewers' comments:

Reviewer #1 (Remarks to the Author):

The manuscript „An affordable high-throughput multispectral imaging system for museum specimens“ by Wei-Ping Chan and colleagues presents a camera-based system that, combined with a high-performance PC system, allows the automated imaging, registration, and calibration of museum drawers of butterflies and extract the critical spectral features from UV to the IR region. The manuscript is well written, very didactic, and presents an interesting new concept. Yet, I think certain parts of the manuscript could be more focused on how to extract new information, especially as the paper is under submission as an article to *Communications Biology*, rather than a methods journal. Beyond that, I will indicate my main issues below, which are all rather minor.

The title promises an “affordable” system, without ever specifying the costs of the used components. Sure, it is built around a standard camera, but I can imagine that the employed filters and particularly the high-performance PC used for data analysis are not particularly “affordable”. I would suggest discussing this point more and giving details on components and costs for all involved materials and give details on the employed computer systems as well as runtimes.

The authors discuss the spectral bands as about 100-200 nm wide (Page 3), but then use narrow line filters instead, and many state-of-the-art hyperspectral systems use bands with a variable bandwidth down to 5-10 nm band widths. I recommend revisiting recent literature and adding this to the manuscript.

How is the UVF different from the UV image without a filter? Both should measure the UV-induced visible light fluorescence, no?

The fact that structure-induced colors (page 7) induce angle-dependency should be supported by references, either already given in the text or broadened to recent literature on this.

Give more details on times to run, how come a 2 min photography series of many species results in a 2-3 hrs runtime per drawer?

L. 193: the low reflectance of *H. glaucippe* is implicitly hidden in the previously presented absorbance spectra of the pterin pigments, so this statement is not fully true.

The whole discussion could use a read with fresh eyes and a particular focus on adding relevant references to claims (i.e., museum collections as repositories, worldwide information exchange, user interface support) to allow a broader view of the fields and their literature support.

Figure 4: top 5 harmonics is not explained in the text or the caption, please add an explanation.

Reviewer #2 (Remarks to the Author):

This is a very ambitious work and has a potential to revolutionize the phenotypic data mining from museum collections. As authors note, museum samples have a great unexplored potential, which could be investigated using sophisticated automatized camera methods that have become recently available. The work describes an entirely new workflow for quantifying museum specimens and especially Lepidoptera. Authors present a construction of the new multispectral imaging rig and describe how to obtain phenotypic data through the workflow. Overall, I find the paper very well written and paving the way into direction to which exploration of digitized museum collections should go in future.

Consequentially, the work is fully focused into describing the new system and I felt that this paper could be better suited to a more method-oriented journal than Communications Biology. But then again, Communications Biology provides a forum for high-quality research regardless of the approach. Certainly, this work would provoke high intrigue in the community. One of the downsides with the current journal format is that this is a methods paper, but as the methods are placed at the very end according to instructions, which makes it more difficult to unpack the complicated method. Authors circumvent this by placing certain sections such as 'the imaging system design' into results in order to improve the flow, but these results are really about describing the method. I understand this choice however and I do not think it is a major issue.

I do not have major comments as regards to the manuscript. One minor thing that would deserve more emphasis is that how the workflow extracts names for different individuals in the dataframe. I see method can handle a museum drawer of specimens in one go and then it is possible to extract data for each specimen in the drawer as the workflow uses fNIR wavelengths to separate them from the box. But what then, does the software give a running number for all the specimens or is it somewhat semi-automatic, or how flexible manual options are what comes to data handling. What if the drawer includes specimen of different types of Lepidoptera, as in the illustrated images; would it be best if there is only one type of specimen group in the drawer, which would allow better data handling? Either way, I found the paper very engaging and hope all the best with it.

Response to Comments on “An affordable high-throughput multispectral imaging system for museum specimens”

Authors: Wei-Ping Chan^{1,2†*}, Richard Rabideau Childers^{1,2†}, Sorcha Ashe^{1†}, Cheng-Chia Tsai³, Caroline Elson¹, Kirsten J. Keleher^{4,5}, Rachel L. Hawkins Sipe², Crystal A. Maier², Lawrence F. Gall⁶, Gary D. Bernard⁷, Edward R. Soucy⁸, Nanfang Yu³, and Naomi E. Pierce^{1,2*}

Affiliations:

¹ Department of Organismic and Evolutionary Biology, Harvard University, Cambridge, MA 02138

² Museum of Comparative Zoology, Harvard University, Cambridge, MA 02138

³ Department of Applied Physics and Applied Mathematics, Columbia University, New York, NY10027

⁴ Department of Forestry and Environmental Resources, North Carolina State University, Raleigh, NC 27695

⁵ Department of Neurobiology and Behavior, Cornell University, Ithaca, NY 14853

⁶ Computer Systems Office & Division of Entomology, Peabody Museum of Natural History, Yale University, New Haven, CT 06520

⁷ Department of Electrical and Computer Engineering, University of Washington, Seattle, WA 98195

⁸ Center for Brain Science, Harvard University, 52 Oxford St. Room 331, Cambridge, MA 02138

†These authors contributed equally to this work

*Correspondence to: chanw@g.harvard.edu, npierce@oeb.harvard.edu

Thank you for the comments and suggestions from the two reviewers. We found them very helpful and revised our manuscript accordingly. Please find the point-to-point responses below.

Reviewer #1 (Remarks to the Author):

The manuscript „An affordable high-throughput multispectral imaging system for museum specimens” by Wei-Ping Chan and colleagues presents a camera-based system that, combined with a high-performance PC system, allows the automated imaging, registration, and calibration of museum drawers of butterflies and extract the critical spectral features from UV to the IR region. The manuscript is well written, very didactic, and presents an interesting new concept. Yet, I think certain parts of the manuscript could be more focused on how to extract new information, especially as the paper is under submission as an article to Communications Biology, rather than a methods journal. Beyond that, I will indicate my main issues below, which are all rather minor.

Comment 1:

The title promises an “affordable” system, without ever specifying the costs of the used components. Sure, it is built around a standard camera, but I can imagine that the employed filters and particularly the high-performance PC used for data analysis are not particularly “affordable”. I would suggest discussing this point more and giving details on components and costs for all involved materials and give details on the employed computer systems as well as runtimes.

Author response:

Thank you for the suggestion. We added a paragraph indicating the detailed hardware and software costs (L131-132, L149-151, L560 - 584).

Comment 2:

The authors discuss the spectral bands as about 100-200 nm wide (Page 3), but then use narrow line filters instead, and many state-of-the-art hyperspectral systems use bands with a variable bandwidth down to 5-10 nm band widths. I recommend revisiting recent literature and adding this to the manuscript.

Author response:

Thank you for pointing this out. We checked with some imaging systems and revised our manuscript to include the state-of-the-art systems (L67 - 68). “Some state-of-the-art imaging systems have 10-20 times finer spectral resolution (~5-10 nm), yet cost 650 times more (~\$350,000).”

Comment 3:

How is the UVF different from the UV image without a filter? Both should measure the UV-induced visible light fluorescence, no?

Author response:

Thank you for the question. We checked our description and realized that our text was misleading. We therefore edited the corresponding section for better clarity (L137 - 139). "... UV-only ($\lambda=360$ nm; reflected light filtered through a Hoya U-340 UV-pass filter on the camera; combined UV reflectance and unfiltered visible fluorescence (called UVF hereafter) comprised of both reflected UV and all UV-induced visible fluorescence; ..."

The normal camera consists of a high-resolution SLR camera (Nikon D800) with its internal UV-IR filter, so only visible light will be detected by the sensor. We removed this filter to allow for UV-visible-IR imaging. Shined by UV light, a range of spectrum (UV reflectance and visible fluorescence) reflect from object surface. We used an UV filter to capture the UV-only reflectance ($\lambda=360$ nm; reflected light filtered through a Hoya U-340 UV-pass filter on the camera, so only reflectance in UV band can be detected by the sensor).

In our system, the full reflected spectrum is called UVF: UV reflectance and visible fluorescence. Under UV illumination, both reflected UV light and the UV induced visible fluorescence were detected without a filter on the camera.

Comment 4:

The fact that structure-induced colors (page 7) induce angle-dependency should be supported by references, either already given in the text or broadened to recent literature on this.

Author response:

Thank you for pointing this out. We added a citation according to your suggestion (L175). "... suggesting whether additional studies should be carried out to investigate polarization at other viewing or incident light angles³³."

Comment 5:

Give more details on times to run, how come a 2 min photography series of many species results in a 2-3 hrs runtime per drawer?

Author response:

Thank you for pointing this out. We added more description in the manuscript to make this clear (L396 - 401). We tried to maximize the degree of automation, so multiple algorithms were used simultaneously, which turns a task from minutes to hours with better generalization for highly varied sets of specimens. Though this takes computational time, it reduces time wasted on downstream manual handling of outliers not suited to a simple one-size-fits-all algorithm, which is even more time consuming under high-throughput conditions. In short, we chose to let the computer work harder and longer to reduce human labor.

Comment 6:

L. 193: the low reflectance of *H. glaucippe* is implicitly hidden in the previously presented absorbance spectra of the pterin pigments, so this statement is not fully true.

Author response:

We edited the sentence as the reviewer suggested (L195 - 196). "Similarly, the white background on *Hebomoia glaucippe* shows little UV reflectance¹⁴,"

Comment 7:

The whole discussion could use a read with fresh eyes and a particular focus on adding relevant references to claims (i.e., museum collections as repositories, worldwide information exchange, user interface support) to allow a broader view of the fields and their literature support.

Author response:

Thank you for the suggestion. We added a section to discuss the digitalization of museum collections and the user interface (L305 - 316).

Comment 8:

Figure 4: top 5 harmonics is not explained in the text or the caption, please add an explanation.

Author response:

We added the information in the caption as the reviewer suggested (Fig. 4; L719 - 721). “In the left panel, the red boundary represents the reconstructed rough shape of a hindwing based on the top five harmonics after being projected into the frequency domain by elliptical Fourier analysis.”

Reviewer #2 (Remarks to the Author):

This is a very ambitious work and has a potential to revolutionize the phenotypic data mining from museum collections. As authors note, museum samples have a great unexplored potential, which could be investigated using sophisticated automatized camera methods that have become recently available. The work describes an entirely new workflow for quantifying museum specimens and especially Lepidoptera. Authors present a construction of the new multispectral imaging rig and describe how to obtain phenotypic data through the workflow. Overall, I find the paper very well written and paving the way into direction to which exploration of digitized museum collections should go in future.

Consequentially, the work is fully focused into describing the new system and I felt that this paper could be better suited to a more method-oriented journal than Communications Biology. But then again, Communications Biology provides a forum for high-quality research regardless of the approach. Certainly, this work would provoke high intrigue in the community. One of the downsides with the current journal format is that this is a methods paper, but as the methods are placed at the very end according to instructions, which makes it more difficult to unpack the complicated method. Authors circumvent this by placing certain sections such as ‘the imaging system design’ into results in order to improve the flow, but these results are really about describing the method. I understand this choice however and I do not think it is a major issue.

Comment 1:

I do not have major comments as regards to the manuscript. One minor thing that would deserve more emphasis is that how the workflow extracts names for different individuals in the dataframe. I see method can handle a museum drawer of specimens in one go and then it is possible to extract data for each specimen in the drawer as the workflow uses fNIR wavelengths to separate them from the box. But what then, does the software give a running number for all the specimens or is it somewhat semi-automatic, or how flexible manual options are what comes

to data handling. What if the drawer includes specimen of different types of Lepidoptera, as in the illustrated images; would it be best if there is only one type of specimen group in the drawer, which would allow better data handling? Either way, I found the paper very engaging and hope all the best with it.

Author response:

Thank you for your suggestion. We added a section to address the details about barcode handling and its flexibility (L483 - 487). We also have a detailed description about it in our online Protocol. Another section mentioning the flexibility of specimen types is also added (L446 - 447). “Given that our approach was designed to accommodate the high diversity in Lepidoptera, specimens with diverse wing shapes can be imaged together (Fig. 3).”

REVIEWERS' COMMENTS:

Reviewer #1 (Remarks to the Author):

This is a nice revision that adequately incorporated most suggestions. Just one small edit: a 350 kUSD piece of equipment is not 650 times more expensive than your system, but rather about 70 times so with the numbers you give. Please correct. I also suggest to add a component list in the SI- including links to representative sellers, this will tremendously help those interested to adapt your technique.

Responses to Reviewers' Comments on "A high-throughput multispectral imaging system for museum specimens"

Authors: Wei-Ping Chan^{1,2†*}, Richard Rabideau Childers^{1,2†}, Sorcha Ashe^{1†}, Cheng-Chia Tsai³, Caroline Elson¹, Kirsten J. Keleher^{4,5}, Rachel L. Hawkins Sipe², Crystal A. Maier², Andrei Sourakov⁶, Lawrence F. Gall⁷, Gary D. Bernard⁸, Edward R. Soucy⁹, Nanfang Yu³, and Naomi E. Pierce^{1,2*}

Editor's Comment:

Your manuscript entitled "An affordable high-throughput multispectral imaging system for museum specimens" has now been seen again by our referees, whose comments appear below. In light of their advice I am delighted to say that we are happy, in principle, to publish a suitably revised version in *Communications Biology* under the open access CC BY license (Creative Commons Attribution v4.0 International License).

We are excited that *Communications Biology* is interested in publishing our work, and we have done our best to revise our manuscript. Please do not hesitate to contact us for further clarifications or revisions.

We do not suggest including a sellers list as the reviewer mentions, but do permit the inclusion of a suggested parts/component list if you think this would be useful for readers.

Given that a sellers list is not encouraged, and the components are generally listed together in the section 'Imaging system design', we did not add an additional component list as the reviewer mentioned.

We therefore invite you to revise your paper one last time to address the remaining concerns of our reviewers. At the same time we ask that you edit your manuscript to comply with our format requirements and to maximise the accessibility and therefore the impact of your work.

Please note that it may still be possible for your paper to be published before the end of 2022, but in order to do this we will need you to address these points as quickly as possible so that we can move forward with your paper.

Thank you. If possible, we would very much like for our paper to be published before the end of 2022, so we prepared this final revision as quickly as we could.

* Please see the attached document for editorial requests for the final version (.docx file). Please ensure a completed version of this file is uploaded as a Related Manuscript with your final submission.

* Please review our final submission file checklist to ensure all necessary files are present with your final submission and to avoid delays in accepting your manuscript. For your reference, a style and formatting guide is available here and includes all of our style requirements.

* An updated editorial policy checklist that verifies compliance with all required editorial policies must be completed and uploaded with the revised manuscript. All points on the policy checklist must be addressed; if needed, please revise your manuscript in response to these points. Please note that this form is a dynamic 'smart pdf' and must therefore be downloaded and completed in Adobe Reader. <https://www.nature.com/documents/nr-editorial-policy-checklist.pdf>

These requirements were all carefully handled in the final revision.

Reviewer #1 (Remarks to the Author):

This is a nice revision that adequately incorporated most suggestions. Just one small edit: a 350 kUSD piece of equipment is not 650 times more expensive than your system, but rather about 70 times so with the numbers you give. Please correct.

Thank you for pointing this out. We have corrected the number. (L85)

I also suggest to add a component list in the SI- including links to representative sellers, this will tremendously help those interested to adapt your technique.

Given that a sellers list is not encouraged by the editor, and the components are generally listed together in the section 'Imaging system design', we did not add an additional component list in the SI.